# Pyridine-based strategies towards nitrogen isotope exchange and multiple isotope incorporation

Minghao Feng [1], Maylis Norlöff[1], Benoit Guichard[1], Steven Kealey[2], Timothée D'Anfray[1], Pierre Thuéry[3], Frédéric Taran[1], Antony Gee [2], Sophie Feuillastre [1] ✉ & Davide Audisio [1] ✉

Isotopic labeling is at the core of health and life science applications such as nuclear imaging, metabolomics and plays a central role in drug development. The rapid access to isotopically labeled organic molecules is a sine qua non condition to support these societally vital areas of research. Based on a rationally driven approach, this study presents an innovative solution to access labeled pyridines by a nitrogen isotope exchange reaction based on a Zincke activation strategy. The technology conceptualizes an opportunity in the field of isotope labeling. $^{15}$N-labeling of pyridines and other relevant heterocycles such as pyrimidines and isoquinolines showcases on a large set of derivatives, including pharmaceuticals. Finally, we explore a nitrogen-to-carbon exchange strategy in order to access $^{13}$C-labeled phenyl derivatives and deuterium labeling of mono-substituted benzene from pyridine-$^2$H$_5$. These results open alternative avenues for multiple isotope labeling on aromatic cores.

Isotope labeling is of paramount importance in diverse areas constituting a multi-billion dollar global market including drugs, agrochemicals, diagnostics and smart materials[1–6]. In contrast with classical organic synthesis, isotope chemists must drift toward unusual hurdles and face specific constraints, such as the high costs of building blocks, the narrow cohort of starting materials available and the constraints imposed by working with radioactivity, often within challenging time frames.

Among nitrogen-based heterocycles, pyridine derivatives represent the ultimate biologically active scaffold (Fig. 1a). In 2014, Njardarson and co-workers have reported that pyridine is the second most common nitrogen heterocycle in U.S. FDA approved drugs[7] and many agrochemicals contain this core structure[8]. Not surprisingly, radiochemists have been interested in labeling such an ubiquitous pharmacophore. In its essence, the pyridine moiety is composed of three elements: carbon, hydrogen and nitrogen (Fig. 1b). Hydrogen isotope labeling is one of the most common technology and allows the

insertion of deuterium and tritium (β$^-$ emitter, T$_{1/2}$ 12.43 years) into pyridine scaffolds[9]. Hydrogen isotope exchange (HIE) using tritium has proven to be the most straightforward technology for pyridine labeling in a single radioactive step[3,4,10–14]. In stark contrast, carbon and nitrogen isotope labeling of pyridines remains challenging, as they are located at the core of the heterocycle. Carbon labeling with $^{13}$C and $^{14}$C (β$^-$ emitter, T$_{1/2}$ 5730 years) can be a tedious multi-step process based on archaic strategies, which do not meet the stringent efficiency requirements of our current society[15–20]. To the best of our knowledge, there are no reports on the core labeling of pyridines with the short-lived positron emitter $^{11}$C (T$_{1/2}$ 20.4 min).

Nitrogen isotope labeling of pyridine bears great promise. Two stable isotopes of nitrogen exist, $^{14}$N and $^{15}$N, with a natural abundance of 99.636% and 0.364%, respectively[21]. Radioactive isotopes of this element are known, but only nitrogen-13 ($^{13}$N, β$^+$ emitter, T$_{1/2}$ 9.97 min) allows for applications in designing innovative radiotracers in positron emission tomography (PET). Unfortunately, due to its challenging

---

[1]Université Paris-Saclay, CEA, Service de Chimie Bio-organique et Marquage, DMTS, F-91191 Gif-sur-Yvette, France. [2]King's College London, School of Biomedical Engineering and Imaging Sciences, Department of Imaging Chemistry and Biology, 4th Floor Lambeth Wing, St Thomas' Hospital, London SE1 7EH, UK. [3]Université Paris-Saclay, CEA, CNRS, NIMBE, 91191 Gif-sur-Yvette, France. ✉e-mail: sophie.feuillastre@cea.fr; davide.audisio@cea.fr

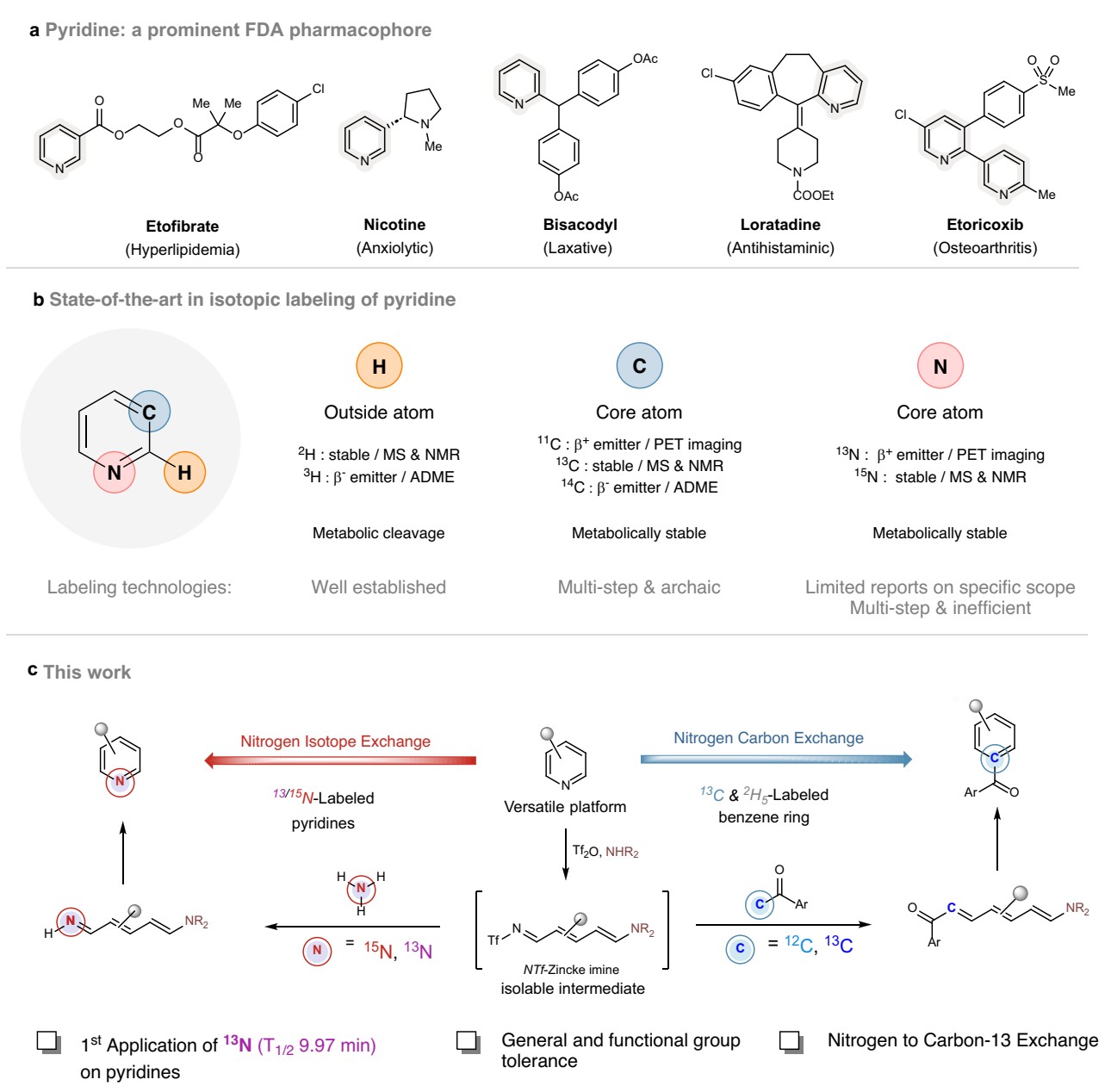

**a Pyridine: a prominent FDA pharmacophore**

**Etofibrate**
(Hyperlipidemia)

**Nicotine**
(Anxiolytic)

**Bisacodyl**
(Laxative)

**Loratadine**
(Antihistaminic)

**Etoricoxib**
(Osteoarthritis)

**b State-of-the-art in isotopic labeling of pyridine**

H
Outside atom

$^2H$ : stable / MS & NMR
$^3H$ : β⁻ emitter / ADME

Metabolic cleavage

C
Core atom

$^{11}C$ : β⁺ emitter / PET imaging
$^{13}C$ : stable / MS & NMR
$^{14}C$ : β⁻ emitter / ADME

Metabolically stable

N
Core atom

$^{13}N$ : β⁺ emitter / PET imaging
$^{15}N$ : stable / MS & NMR

Metabolically stable

Labeling technologies:     Well established     Multi-step & archaic     Limited reports on specific scope
Multi-step & inefficient

**c This work**

Nitrogen Isotope Exchange

$^{13/15}N$-Labeled pyridines

Versatile platform

Nitrogen Carbon Exchange

$^{13}C$ & $^2H_5$-Labeled benzene ring

Tf₂O, NHR₂

$N = ^{15}N, ^{13}N$

*NTf*-Zincke imine
isolable intermediate

$C = ^{12}C, ^{13}C$

☐ 1st Application of $^{13}N$ (T$_{1/2}$ 9.97 min) on pyridines

☐ $^{15}NH_3$ or $^{13}NH_3$ primary isotopic sources

☐ General and functional group tolerance

☐ Simple and mild procedure

☐ Nitrogen to Carbon-13 Exchange

☐ Deuterium Labeling

**Fig. 1 | Isotope exchange reactions of pyridine. a** Pyridine: a prominent FDA pharmacophore. **b** State-of-the-art in isotopic labeling of pyridine. **c** This strategy: nitrogen isotope exchange and nitrogen-to-carbon replacement for benzene labeling with carbon-13 and deuterium.

short half-life, requiring its clinical application within minutes of tracer synthesis, $^{11}C$ or $^{18}F$ counterparts have often been preferred, drastically reducing the $^{13}N$-radiolabeling state of the art[22–25].

In order to bridge these gaps and based on our previous experience on development of isotopic exchange approaches to achieve late-stage-labeling of biologically active molecules[26–29], we have investigated a pyridine-based scaffold isotopic editing technology allowing nitrogen isotope exchange (NIE), using labeled ammonia as the primary isotopic nitrogen source (Fig. 1c, left)[30,31].

The replacement of naturally abundant $^{14}N$ by its isotopes (i.e. NIE) is an under-developed concept. Specific examples exist, but are substrate specific, limited in scope, require two-step processes and imply a structural modification between the starting material and the labeled product[32–34]. To the best of our knowledge, direct NIE without structural modifications is limited to nitrile metathesis in presence of

molybdenum and tungsten catalysts[35] and to one single example of a primary sulfonamide[36].

Labeling of pyridine by NIE was first reported by Oppenheimer et al. in 1978. This procedure is based on the reaction between a Zincke pyridinium salt and labeled [$^{15}N$]NH₄Cl[37]. While potentially appealing, the reliability of this transformation seemed questionable, as further implementation showcased a narrow synthetic application[38–41]. In addition, a series of pitfalls were identified: i) the need for long reaction times, incompatible with short-lived $^{13}N$; ii) the inherent two-step nature of the procedure; iii) the well-established intolerance of the Zincke reaction towards pyridine substitution[42,43]. After preliminary investigations (see Supplementary Information in Section 2, page S5), we confirmed the narrow synthetic scope of the Zincke strategy and decided to move towards a more convenient approach[44]. Activation of the pyridine core in presence of

trifluoromethanesulfonic anhydride (Tf$_2$O) has demonstrated high potential for their late-stage functionalization under mild conditions[45–47]. In 1997, Toscano et al.[48] showed that triflypyridinium triflate (TPT) can be prepared in situ and underwent a ring-opening process in presence of amines to form the corresponding conjugated iminium species (NTf-Zincke imines)[49]. In 2022, Paton and McNally reported an application of this reaction concept in performing the otherwise challenging halogenation of the 3-position[50]. The same year, Sarpong and co-workers published a skeletal editing of pyrimidine with Tf$_2$O activation to access pyrazoles[51]. Leveraging of such strategies, we sought to achieve NIE on pyridine moieties via NTf-Zincke imines. Aiming to provide an entry toward [13]N-labeling of this pharmaceutically relevant scaffold, we recognized the mandatory use of labeled ammonia as fundamental to implement such an ambitious goal.

The results presented herein show that, by rational design and careful reaction optimization, isotope replacement can be achieved. This technology was applied to the stable [15]N-labeling of substituted pyridines, isoquinolines and pyrimidines, including pharmaceuticals. In addition, we provide a proof-of-concept [13]N-labeling of pyridines in one single radioactive step from the universal precursor [[13]N]NH$_3$. Finally, taking full advantage of the pyridyl platform, we have explored nitrogen-to-carbon exchange to access [13]C-labeled phenyl derivatives and multi-labeled mono-substituted [[13]C, [2]H$_5$]benzene starting from commercially available pyridine-[2]H$_5$ (Fig. 1c, right).

## Results and discussion

### Reaction design and optimization

To validate our hypothesis, 2-phenylpyridine **1** was selected as a model substrate for testing the [14]N/[15]N exchange. As anticipated, 2-phenylpyridine **1** was converted to the corresponding NTf-Zincke imine intermediate **Im1** after the nucleophilic attack of dibenzylamine (Table 1, Entry 1) on the TPT. [[15]N]NH$_4$Cl, one of the most readily available [15]N sources, was employed to react with the NTf-Zincke imine intermediate **Im1** in presence of triethylamine to generate [[15]N]NH$_3$

in situ. After subsequent cyclization, [15]N-labeled 2-phenylpyridine **[[15]N] 1** was afforded in 99% yield with 68% [15]N isotopic enrichment (IE, meaning that 68% of the isolated product now bears a [15]N atom). The competition between the nucleophilic attack of the dibenzylamine at the C$_2$ position of the Tf-pyridine intermediate and the sulfur atom of the triflic moiety could result in the desired NTf-Zincke imine intermediate **Im1** and the 2-phenylpyridine **1** respectively, the latter being considered as the unlabeled component in the product (see Supplementary Information, Section 4 for detailed proposed mechanism). Increasing the efficiency of the conversion from starting material to the NTf-Zincke imine intermediate was considered to be a key factor to further improve the IE. A higher temperature for the nucleophilic attack of the amine on the activated pyridine was found to be beneficial to product IE (Entry 2-3). At 60 °C, the desired product was afforded in 87% yield with 76% IE. A variety of nitrogen-bearing nucleophiles were screened in the ring-opening step of 2-phenylpyridine since the amine substitution can affect their nucleophilicity as well as the stability of the afforded NTf-Zincke imines. Representative results are illustrated in Table 1, entries 4-8 (see the Supplementary Information for the full screening tables in Section 3, Supplementary Tables 2 and 3, pages S6, S7). When tetrahydroquinoline was used, only 42% IE was obtained for the labeled product (Entry 4). Nonetheless, 71% IE was afforded when indoline was involved (Entry 5). A series of tests with indolines bearing different substituents were not fruitful (see Supplementary Information for details in Section 3, Supplementary Table 3, page S7). N-Methylaniline was found less effective than dibenzylamine (Entry 6). Less sterically hindered nucleophiles, such as diethylamine (Entry 7) and methylamine (Entry 8) resulted in unidentified by-products and a low yield of unlabeled product. Reasoning towards a full isotope replacement, we were pleased to observe that by precipitating the NTf-Zincke imine intermediate **Im1** in n-hexane, the unlabeled 2-phenylpyridine **1** could be easily removed from the mixture. The precipitated NTf-Zincke imine intermediate **Im1** was used in the subsequent labeling stage without further purification, yielding 97% of [15]N-enriched 2-phenylpyridine

**Table 1 | Representative conditions for the optimization of the [15]N-pyridine labeling[a]**

| Entry | Activation temperature | Nucleophiles | Isotopic enrichment[b] | Yield[c] |
|---|---|---|---|---|
| 1[d] | −78 to 25 °C | dibenzylamine | 68% | 99% |
| 2[d] | 40 °C | dibenzylamine | 71% | 88% |
| 3 | 60 °C | dibenzylamine | 76% | 87% |
| 4 | 60 °C | tetrahydroquinoline | 42% | 99% |
| 5 | 60 °C | indoline | 71% | 99% |
| 6 | 60 °C | N-methylaniline | 62% | 97% |
| 7 | 60 °C | diethylamine | <5% | 51% |
| 8 | 60 °C | methylamine | <5% | 65% |
| 9[e] | 60 °C | dibenzylamine | 97% | 64%[f] |

Reaction conditions: [a] 2-Phenylpyridine (0.2 mmol, 1.0 equiv.), Tf$_2$O (0.2 mmol, 1.0 equiv.), ethyl acetate (1.0 mL), −78 °C, 0.5 h, then nucleophiles (0.24 mmol, 1.2 equiv.), 2,4,6-collidine (0.2 mmol, 1.0 equiv.), indicated temperature, 1.0 h, then [15]NH$_4$Cl (0.6 mmol, 3.0 equiv.), triethylamine (1.2 mmol, 6.0 equiv.), acetonitrile (2.0 mL), 100 °C, 1 h. Isotopic enrichment (IE) expresses the percentage of [15]N in the isolated product.
[b]Measured by LC-MS.
[c][1]H-NMR yields using dibromomethane as an internal standard.
[d]Dichloromethane was used instead of ethyl acetate.
[e]Imine intermediate was isolated by precipitation and then used in the labeling step.
[f]Combined two-step yield of isolated product [[15]N]**1**.

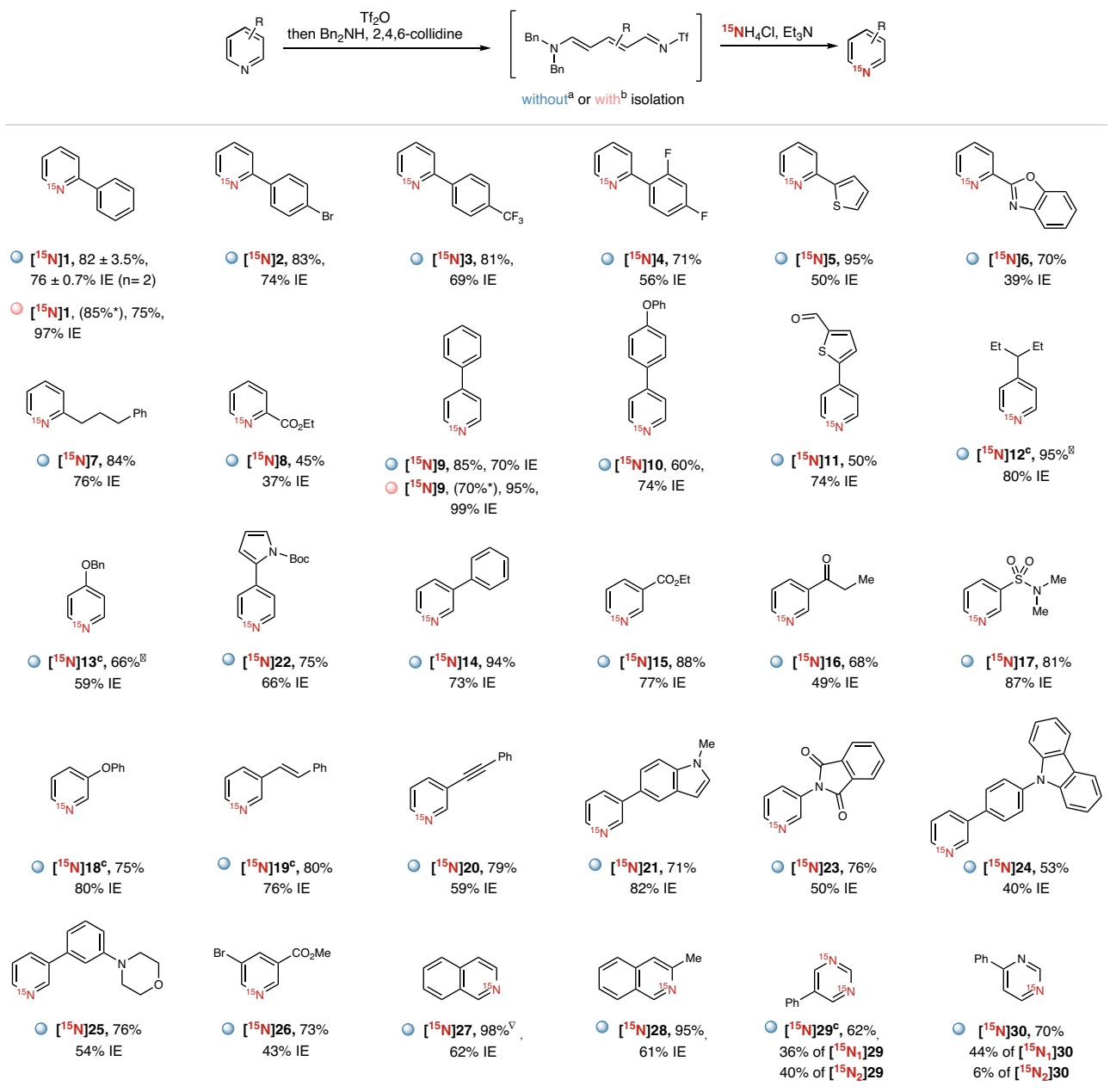

**Fig. 2 | ¹⁵N-Labeling substrate scope.** Reaction conditions: [a]Pyridines (0.2 mmol, 1.0 equiv.), Tf₂O (0.2 mmol, 1.0 equiv.), ethyl acetate (1.0 mL), −78 °C, 0.5 h, then dibenzylamine (0.24 mmol, 1.2 equiv.), 2,4,6-collidine (0.2 mmol, 1.0 equiv.), 60 °C, 1.0 h, then ¹⁵NH₄Cl (0.6 mmol, 3.0 equiv.), triethylamine (1.2 mmol, 6.0 equiv.), acetonitrile (2.0 mL), 100 °C, 1 h. Yields of isolated products are shown. Isotopic enrichment (IE) expresses the percentage of ¹⁵N in the isolated product and was determined by HRMS. [b]Imine intermediate was isolated by precipitation and then used in the labeling step. *Yields of isolated imines. ▽¹H-NMR yields using dibromomethane as an internal standard. [c]Dichloromethane was used instead of ethyl acetate and the reaction temperature was 25 °C in the first step.

[¹⁵N]1 in 64% overall yield. This observation was a keystone for the development of our ¹³N strategy, where isotope dilution is unsuitable, due to the limited scale of production of the radionuclide.

## Reaction scope

With the optimized conditions in hand, we explored the scope of this one-pot pyridine-based nitrogen isotope exchange strategy. Benefitting from the excellent chemoselectivity of the Tf₂O activation, this transformation displayed excellent functional group tolerance. Pyridines bearing substituents at positions 2, 3 and 4 could be efficiently converted to their corresponding ¹⁵N-labeled counterparts. In contrast to the traditional Zincke strategy, this procedure is fully compatible with a

variety of substituents at position 2 to give the corresponding ¹⁵N-labeled pyridines (Fig. 2, [¹⁵N]1 to [¹⁵N]8) with moderate to good IE. Due to the steric hindrance, 2-substituted pyridines cannot afford the traditional Zincke salts when reacting with 1-chloro-2,4-dinitrobenzene (see Supplementary Information for details in Section 2, page S5). Heterocyclic moieties such as thiophene ([¹⁵N]5), benzo[d]oxazole ([¹⁵N]6) substituted pyridines are tolerated in this transformation. However, 2-(pyridin-2-yl)benzo[d]oxazole 6 afforded the product with a lower IE (39%) since benzo[d]oxazole might react with Tf₂O. 2-Alkylated pyridine gave product [¹⁵N]7 with 76% IE. The ester moiety at position 2 of pyridine is tolerated, as well ([¹⁵N]8). Pyridines with 4-substitued alkyl and aryl moieties are effective substrates ([¹⁵N]9-13, 22). The electron

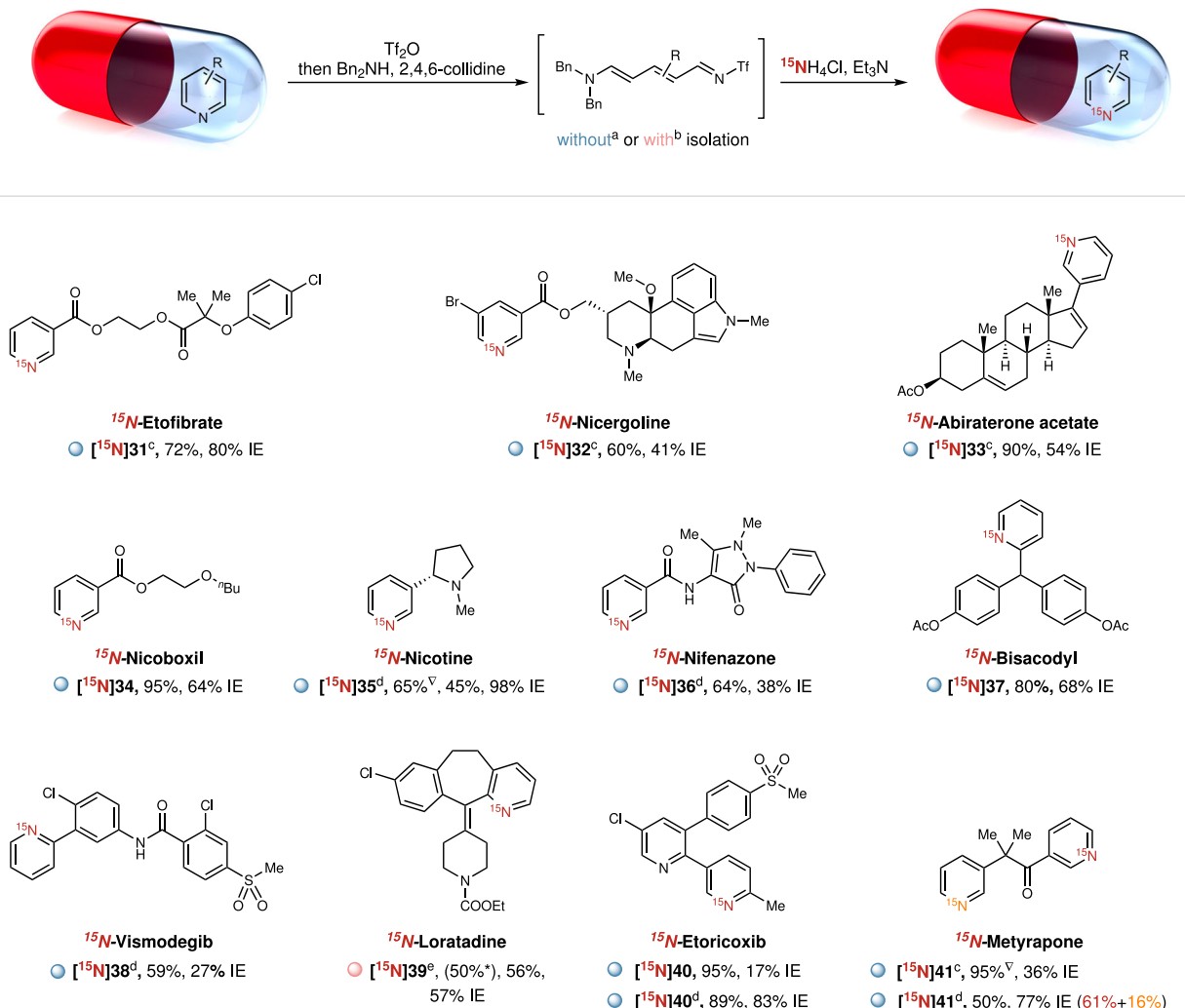

**Fig. 3 | 15N-Labeling of pharmaceutical molecules.** Reaction conditions: [a]Pyridines (0.1–0.2 mmol, 1.0 equiv.), Tf$_2$O (1.0 equiv.), ethyl acetate (0.5–1.0 mL), −78 °C, 0.5 h, then dibenzylamine (0.24 mmol, 1.2 equiv.), 2,4,6-collidine (1.0 equiv.), 60 °C, 1.0 h, then 15NH$_4$Cl (3.0 equiv.), triethylamine (6.0 equiv.), acetonitrile (1.0–2.0 mL), 100 °C, 1 h. Yields of isolated products are shown. Isotopic enrichment (IE) expresses the percentage of 15N in the isolated product and was determined by HRMS. Modified procedures: [b]Imine intermediate was isolated by precipitation and then used in the labeling step. *Yields of isolated imines. [c]Dichloromethane was used instead of ethyl acetate in the first step and the reaction temperature was 25 °C. [d]Pyridines (0.2 mmol, 1.0 equiv.), Tf$_2$O (2.0 equiv.), dichloromethane (1.0 mL), −78 °C, 0.5 h, then dibenzylamine (4.8 equiv.), 2,4,6-collidine (2.0 equiv), 25 °C, 1.0 h, then 15NH$_4$Cl (3.0 equiv.), triethylamine (6.0 equiv.), acetonitrile (2.5 mL), 100 °C, 1 h. [e]Reaction stirred for 24 h for the labeling-cyclization step. [▽]1H-NMR yields using dibromomethane as an internal standard.

donating group at position 4 ([15N]13) does not affect the reaction. Notably, the aldehyde moiety is tolerated, delivering the desired 15N-labeled pyridine [15N]11 in 50% yield and 74% IE. Pyridines with a variety of substituents at position 3, including aryl ([15N]14), ester ([15N]15), ketone ([15N]16), sulfonamide ([15N]17), phenoxy group ([15N]18), alkene ([15N]19), alkyne ([15N]20), as well as other heterocycles ([15N]21, 23, 24), proved to be efficient in this one-pot 15N-labeling procedure, affording the labeled pyridines with 40% to 87% IE. To our delight, we could also achieve the 15N-labeling on 3,5-disubstituted pyridine ([15N]26), isoquinolines ([15N]27, [15N]28) and pyrimidines ([15N]29 and [15N]30).

The application to late-stage 15N-labeling of elaborated pyridine-containing pharmaceuticals and biologically active molecules was then realized using the developed protocol. As shown in Fig. 3, a variety of 15N-labeled drugs and biologically active compounds were labeled straightforwardly using the corresponding starting material without the need for any functionalization. When other nucleophilic moieties were present in the same structure, an adapted procedure, with

increased equivalents of Tf$_2$O and dibenzylamine, was applied to afford the product with improved isotopic enrichments ([15N]35, [15N]36, [15N]38, [15N]40, [15N]41). Notably, 15N-enriched nicotine [15N]35 was obtained with 98% IE from unfunctionalized nicotine through this adapted one-pot procedure. Secondary amides, which have the potential to react with Tf$_2$O, are well-tolerated in this transformation ([15N]36, [15N]38). While the standard process gave low isotopic enrichment of loratadine, [15N]39 could be obtained in 57% IE by precipitating the corresponding *N*Tf-Zincke imine intermediate. Etoricoxib, a marketed selective COX-2 inhibitor, contains two pyridine subunits in its structure. The nitrogen of the disubstituted pyridine was selectively labeled without affecting the trisubstituted one ([15N]40). In another example, a 61:16 regioselectivity was observed for the 15N-labeling of Metyrapone, favoring the electron-deficient pyridine scaffold ([15N]41).

To assess the utility of this nitrogen isotopic exchange procedure in the field of PET radiochemistry, we next explored its application to 13N-labeling. By cyclotron mediated proton bombardment of a pure

water target containing trace amounts of ethanol, $[^{13}N]NH_3$ was produced in aqueous solution. As a proof-of-concept, the isolated $N$Tf-Zincke imines (**Im1** and **Im9**) from 2-phenylpyridine **1** and 4-phenylpyridine **9** were treated with cyclotron-produced $[^{13}N]NH_3$ at 100 °C for 6 minutes (Supplementary Information, Section 11, pages S53–S58). The desired $[^{13}N]$2-phenylpyridine $[^{13}N]$**1** and $[^{13}N]$4-phenylpyridine $[^{13}N]$**9** were afforded successfully with moderate radiochemical yields of 20% and 78%, respectively. These preliminary results demonstrate that our nitrogen isotopic exchange strategy can be applied to radioactive $^{13}N$-labeling, and further expands the scope of $^{13}N$-labeling methods.

In light of these results, we recognized the hidden potential of pyridine activation to move beyond nitrogen labeling to further apply it to a formal nitrogen-to-carbon exchange. The insertion of carbon isotopes into the benzene scaffold is the Holy Grail for metabolic stability[52]. The classic method to access carbon-labeled benzene is the trimerization of $[^{13}C]$ and $[^{14}C]$acetylene[53–55]. Next, benzene must be functionalized in a multi-step fashion to more valuable and synthetically useful derivatives[56]. A comprehensive view of mono- and di-labeled arenes with carbon isotopes were reviewed by Herbert and co-workers in 2011[57]. In 2023, Hooker, Levin and co-workers published a solution toward $^{11}C$ and $^{13}C$-labeling of 2-6-disubstituted phenols[58]. Despite these efforts, the access to regio-specific labeled and substituted derivatives remains narrow and restricted to specific patterns. We envisioned a formal nitrogen-carbon exchange on pyridines by using a proper $^{13}C$-carbon nucleophile to afford core-labeled functionalized benzenes. Inspired by the pioneering work on the reaction between carbon nucleophiles and Zincke intermediates of Morofuji and Kano[59], as well as the avenue towards cyanine dyes synthesis[60,61], we identified commercial $^{13}C$-acetophenone as suitable nucleophile to react with the activated pyridines or $N$Tf-Zincke imines and obtain $^{13}C_1$-labeled benzene derivatives.

As shown in Fig. 4a, isolated $N$Tf-Zincke imine **Im9**, which was obtained from 4-phenylpyridine and whose structure was confirmed by X-ray diffraction (see Supplementary Information for the details in Section 12, page S59), smoothly reacted with acetophenone in presence of a base. After subsequent cyclization and aromatization, the desired benzophenone derivative **42** was isolated in 90% yield (Fig. 4b). Notably, the transformation could be performed in a one-pot fashion giving the product in 45% yield. Different acetophenones were tested in the reactions with $N$Tf-Zincke imine **Im9**, electron-rich (**43**), bromo-bearing (**44**) as well as thiophene derived methyl ketones (**45** and **46**) and effectively delivered the formal N-C exchange products in suitable yields. Interestingly, pyridines bearing a substituent at position 3 could be selectively converted into the corresponding product **47**, while no trace of other isomers were observed. Unfortunately, 2-phenyl pyridine failed to provide the desired product. As anticipated, $^{13}C_1$-labeled benzophenones ($[^{13}C]$**42** and $[^{13}C]$**47**) were afforded by using $[^{13}C]$acetophenone as nucleophile. To our delight, the developed N-C exchange protocol could be applied to nicotine, affording the corresponding $^{12}C$ and $^{13}C$ derivatives **51** and $[^{13}C]$**51** (Fig. 4d). The synthesis of this $^{13}C$-labeled derivative would be remarkably challenging by existing synthetic methodologies.

Poly-deuterated molecules are used for routine MS-based quantification in pharmaceutical and agrochemical companies and the insertion of multiple deuterium labels (MDL) into organic molecules has received much attention. While HIE is an effective tool to achieve MDL, isotope incorporation does not fulfill complete deuteration, thus providing complex, inseparable mixtures of isotopomers, namely $[M+1]$, $[M+2]$, $[M+3]$ to $[M+n]$. In this context, precision deuteration, i.e. the surgical insertion of the isotope at regiospecific sites of the organic compound, is still in its infancy. We proposed that the deuterated solvent $[^2H_5]$pyridine would be a suitable reagent to enable selective deuteration of benzophenone derivatives, exclusively at the phenyl ring. As illustrated in Fig. 4b, c, $[^2H_5]$benzophenones $[^2H_5]$**42**, $[^2H_5]$**47** and $[^2H_5]$**50** were successfully isolated from $[^2H_5]$pyridine with only minimal deuterium loss. Moreover, we could synthesize an unprecedented $[^2H_5, {}^{13}C_1]$benzophenone isotopomer starting from $[^{13}C]$acetophenone and $[^2H_5]$pyridine. As shown in Fig. 4c, compared to the unlabeled benzophenone **50**, $[^{13}C]$acetophenone $[^{13}C]$**50** and $[^2H_5, {}^{13}C]$**50** showed distinctive $^{13}C$-NMR spectra. Due to the presence of fully enriched $^{13}C$ in the aromatic ring of $[^{13}C_1]$**50**, a series of $^{13}C$-$^{13}C$ couplings were observed, while for $[^2H_5, {}^{13}C]$**50**, deuterium isotope effects on $^{13}C$-NMR signals and multiplicity are in agreement with the literature[62,63].

Finally, the unique potential of this methodology in the preparation of pharmaceutically relevant internal standards was showcased (Fig. 4d). By reducing $[^2H_5, {}^{13}C]$**50** with sodium borohydride, the corresponding diphenylmethanol $[^2H_5, {}^{13}C]$**52** was obtained. The latter was reported as a key common intermediate in the synthesis of cyclizine and cinnarizine[64,65].

In conclusion, this study presents an innovative solution to access labeled pyridines by NIE based on a Zincke activation strategy. The technology conceptualizes an unexplored opportunity in the field of isotope chemistry and prospects to move forward in the challenging isotope labeling realm. $^{15}N$-labeling of pyridine was possible with up-to-full isotope incorporation into a large variety of N-heterocycles, including pyrimidines and isoquinolines. Using labeled ammonia as primary isotopic source, this method proved to be compatible with late-stage NIE of complex pharmaceutical derivatives and proof-of-concept on the application of this technology toward PET suitable $^{13}N$-labeling was provided. Finally, this strategy was implemented for $^{13}C$-core isotope labeling of mono-substituted benzene derivatives and deuterium labeling. We believe this platform for multiple isotope labeling will provide a unique opportunity for future access to stable labeled $^{15}N$, $^{13}C$, $^2H_5$-pyridyl derivatives and will play a be fundamental role for the future development of $^{13}N$-based PET radiotracers and $^{14}C$-labeled aromatics.

## Methods

### General procedure for $^{15}N$-labeling of pyridines

To an oven-dried 8.00 mL pressure vial equipped with a stirring bar, the heterocycle (0.20 mmol, 1.00 equiv.) and ethyl acetate (1.00 mL) were charged under an argon atmosphere. The mixture was cooled to −78 °C and trifluoromethanesulfonic anhydride (33.6 μL, 0.20 mmol, 1.00 equiv.) was added dropwise. The reaction was stirred for 30 minutes at −78 °C before a solution of dibenzylamine (47.0 μL, 0.24 mmol, 1.20 equiv.) and 2,4,6-collidine (26.3 μL, 0.20 mmol, 1.00 equiv.) in ethyl acetate (0.25 mL) was added dropwise. The cooling bath was then removed and the reaction was warmed to 60 °C. After 1 hour, $^{15}N$-ammonium chloride (32.7 mg, 0.60 mmol, 3.00 equiv.), triethylamine (167.3 μL, 1.20 mmol, 6.00 equiv.) and acetonitrile (2.50 mL) were added. The reaction was stirred at 100 °C for 1 h. After cooling to room temperature, the reaction was diluted with dichloromethane (5.00 mL). The mixture was then washed with saturated aqueous ammonium chloride (5.00 mL). The aqueous phase was then extracted with dichloromethane (3 × 5.00 mL). The combined organic phases were dried over magnesium sulfate. The mixture was filtered and the filtrate was concentrated under vacuum to give the crude product. The crude product was purified with silica gel column chromatography. When the desired product presents the same retention factor as dibenzylamine on TLC, the crude product was dissolved in 1.00 mL of chloroform, di-*tert*-butyl dicarbonate (105 mg, 0.48 mmol, 2.40 equiv.) were added and the mixture was stirred at 25 °C until the complete conversion of dibenzylamine (approximately 4 h). The solution was then concentrated and purified on silica gel column to give the desired product.

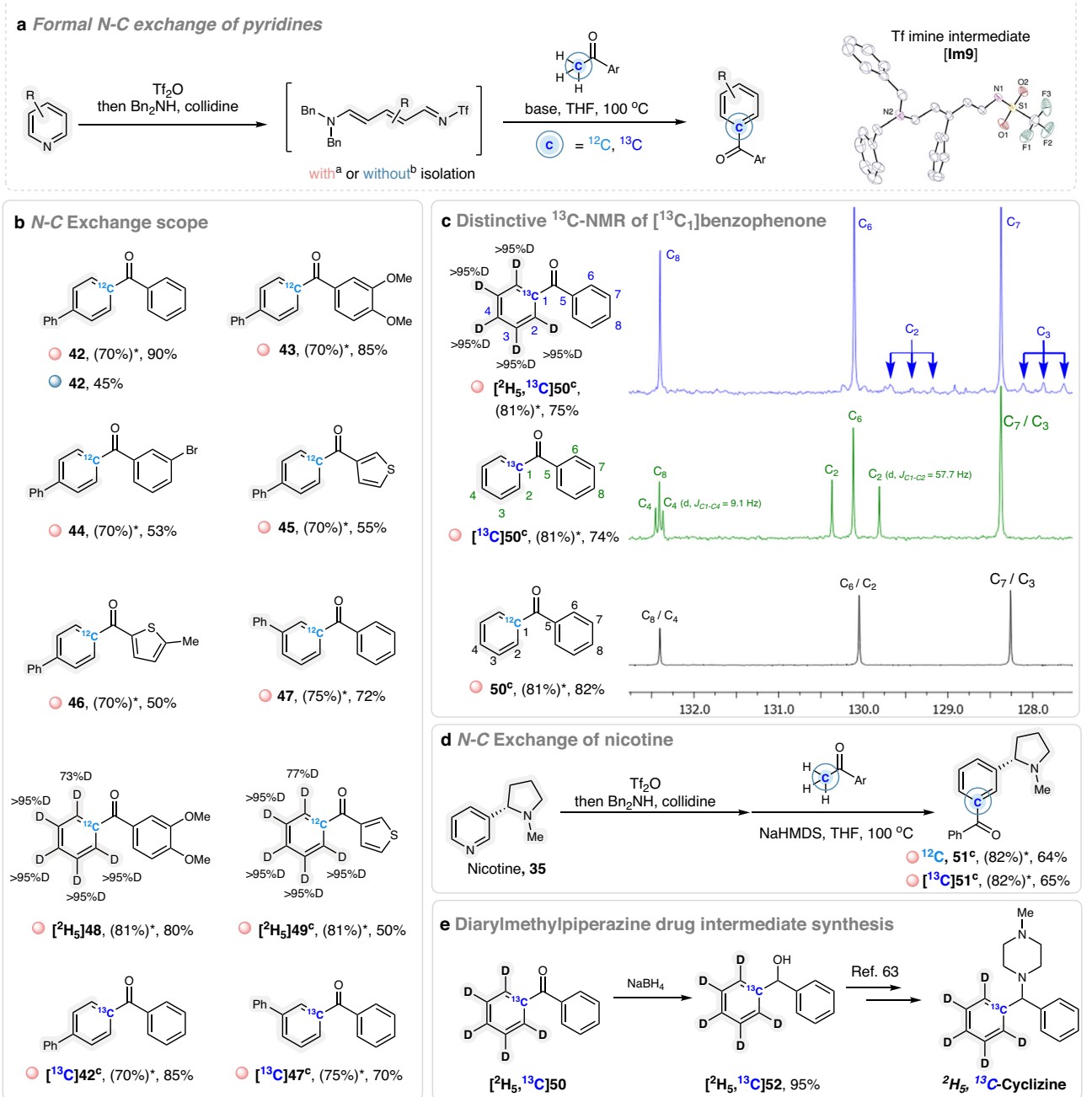

**Fig. 4 | Nitrogen-to-carbon exchange on pyridines and multiple isotope incorporation. a** Formal nitrogen-to-carbon exchange of pyridines and X-ray of Tf*N*-Zincke imine **Im9**. Reaction conditions: [a] Pyridines (0.2 mmol, 2.0 equiv.), Tf₂O (0.2 mmol, 2.0 equiv.), ethyl acetate (1.0 mL), −78 °C, 0.5 h, then dibenzylamine (0.24 mmol, 2.4 equiv.), 2,4,6-collidine (0.2 mmol, 2.0 equiv.), 60 °C, 1.0 h. Imine intermediate was isolated by precipitation. Then ketone (0.1 mmol, 1.0 equiv.), *t*BuONa (0.15 mmol, 1.5 equiv.), THF (2.0 mL), 100 °C, 20 h. **b** N-C exchange scope. Yields of isolated products are shown. [b] Imine intermediate was not isolated. *Yields of isolated imines. Modified reaction conditions: [c] NaHMDS was used instead of *t*BuONa. **c** Distinctive ¹³C-NMR spectra of [¹³C₁]benzophenone (NMR solvent: CDCl₃). **d** Nicotine modification with N-C exchange procedure. **e** [²H₅, ¹³C₁]benzophenone reduction to afford diarylmethylpiperazine drug intermediate.

## Data availability
Experimental procedures and characterization data are provided in the Supplementary Information. Correspondence and requests for materials should be addressed to S.F. and D.A.

## Code availability
CCDC 2304028 (Compound **Im9**) contains the supplementary crystallographic data for this paper. These data can be obtained free of charge via www.ccdc.cam.ac.uk/data_request/cif, or by emailing data_request@ccdc.cam.ac.uk, or by contacting The Cambridge

Crystallographic Data Centre, 12 Union Road, Cambridge CB2 1EZ, UK; fax: +44 1223336033.

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

## Acknowledgements

The authors thank S. Lebrequier and D.-A. Buisson (DRF-JOLIOT-SCBM, CEA) for the excellent analytical support. We thank Dr. G. Pieters (DRF-JOLIOT-SCBM, CEA) for helpful discussion.

## Author contributions

S.F. and D.A. conceived the idea and supervised project. M.F. and D.A. designed the experiments. M.F., M.N., and B.G. performed the experiments, synthesized and characterized the molecules, analyzed the data discussed the results. T.D.A. supervised the whole analytical characterization. F.T. discussed the results and participate to development of the N to C project. S.K. performed the nitrogen-13 labeling. S.K., A.G. analyzed and characterized the nitrogen-13 labeling experiments. P.T. performed the analysis of X-ray diffraction. M.F., S.F. and D.A. prepared the manuscript with contributions from all authors.

## Funding

This project has received funding from CEA and the European Union's Horizon 2020 research and innovation program under grant agreement No. 862179 and the European Research Council (ERC-2019-COG - 864576, D.A.). EPSRC are gratefully acknowledged for the Programme Grant (EP/S032789/1) and Standard Research (EP/W021307/1) funding that supported this work.

## Competing interests

The authors declare no competing interests.
