## [Peer Review File · Nature Communications]

Pyridine-based Strategies towards Nitrogen Isotope Exchange and Multiple Isotope IncorporationEditorial Note: This manuscript has been previously reviewed at another journal that is not operating a transparent peer review scheme. This document only contains reviewer comments and rebuttal letters for versions considered at Nature Communications.

Reviewers' Comments:

Reviewer #2:

Remarks to the Author:

[Note from the Editor: Reviewer #2 was asked to assess the response given to reviewer #1, who was unable to review the revision.]

The authors have addressed most of the concerns of the reviewers. Herein, the reviewer has two questions about points 13 and 33.

Question for point 13: For determining the labeling efficiency, it is acceptable to use radio-HPLC or radio-TLC. However, for labeled compounds that have only been analyzed using HPLC without isolation, it would be more appropriate to describe the labeling efficiency using RCC (radiochemical conversion) instead of RCY (radiochemical yield).

Please review all instances of RCYs in the manuscript and annotate them as either RCCs or non-isolated RCYs (or estimated by radio-HPLC) when the labeled compounds were not isolated for accurate representation of the labeling efficiency.

Question for point 33: The authors have clearly annotated the HRMS data for both the labeled and unlabeled compounds. For the exact mass calculations of $[^{15}\text{N}]_3$ ($\text{C}_{12}\text{H}_9\text{F}_3^{15}\text{N} [\text{M}+\text{H}]^+$), it is important to note that the calculated mass for the positive ion accounts for the mass of the missing electron. Additionally, in mass spectrometry, the term "m/z" refers to the mass-to-charge ratio, where "m" represents the mass of the ion when "z" equals one, rather than the mass of an electrically neutral molecule. Please double-check the HRMS data to ensure accurate representation and interpretation of the mass spectrometry results.

Reviewer #3:

Remarks to the Author:

The authors describe a nitrogen isotope exchange strategy by utilizing a Zincke imine intermediate and extend this method to include carbon isotope enrichment. However, the novelty of this work, also listed as a concern of Reviewer 2, still appears to be slightly diminished, in my opinion, due to the preprints on ChemRxiv highlighting $^{14}\text{N} \rightarrow ^{15}\text{N}$ isotope exchange strategies (doi: 10.26434/chemrxiv-2023-30dtw, doi: 10.26434/chemrxiv-2023-v4hx2, and doi: 10.26434/chemrxiv2023-cb5lg).

Furthermore, the reported pyridine ring opening strategy (Tf₂O 1.0 equiv, HNBn₂ 1.2 equiv, 2,4,6-collidine 1.0 equiv, and EtOAc for 60 °C, 1.0 h), as I also mentioned in my initial review, is extremely similar to the one presented by McNally and coworkers (Tf₂O 1.0 equiv, HNBn₂ 1.2 equiv, collidine 1.0 equiv, and EtOAc for -78 °C, 30 min) in 2022 (10.1126/science.add8980). The work in the other two $^{14}\text{N} \rightarrow ^{15}\text{N}$ preprints (doi: 10.26434/chemrxiv-2023-30dtw, doi and doi: 10.26434/chemrxiv2023-cb5lg), not from McNally and coworkers, at least show a deviation in the ring opening strategy originally reported by McNally, aside from Tf₂O.

The ¹³N-labeling experiment, which is now in the SI, is a novel radiolabeling approach. However, I, as well as the other reviewers, had expressed concerns and questions on this sole example.

Unfortunately, the majority of these concerns were dismissed by authors. As a result, I still believe the

novelty of this body of work does not meet the requirement for Nature Communications. In my opinion, the authors should completely remove the ^{13}N -labeling experiment, and expand on this radiolabeling strategy and scope for a high impact publication. The remaining work should be more suitable for JACS which is where all the above preprints (doi: 10.26434/chemrxiv-2023-30dtw, doi: 10.26434/chemrxiv-2023-v4hx2, and doi: 10.26434/chemrxiv2023-cb5lg) were ultimately published.

Answers to reviewers comments

Reviewer #2 (Remarks to the Author):

[Note from the Editor: Reviewer #2 was asked to assess the response given to reviewer #1, who was unable to review the revision.]

The authors have addressed most of the concerns of the reviewers. Herein, the reviewer has two questions about points 13 and 33.

Question for point 13: For determining the labeling efficiency, it is acceptable to use radio-HPLC or radio-TLC. However, for labeled compounds that have only been analyzed using HPLC without isolation, it would be more appropriate to describe the labeling efficiency using RCC (radiochemical conversion) instead of RCY (radiochemical yield).

Please review all instances of RCYs in the manuscript and annotate them as either RCCs or non-isolated RCYs (or estimated by radio-HPLC) when the labeled compounds were not isolated for accurate representation of the labeling efficiency.

Answer :

As a reminder of our previous revision/improvements of the manuscript, these preliminary results were moved from the manuscript to the supporting information.

As suggested by reviewer #2, we have modified RCYs with non-isolated RCYs in the SI (pages S51-S53) for the two ¹³N labelling experiments.

Modifications were implemented with colours.

Question for point 33: The authors have clearly annotated the HRMS data for both the labeled and unlabeled compounds. For the exact mass calculations of [15N]3 (C₁₂H₉F₃¹⁵N [M+H]⁺), it is important to note that the calculated mass for the positive ion accounts for the mass of the missing electron. Additionally, in mass spectrometry, the term "m/z" refers to the mass-to-charge ratio, where "m" represents the mass of the ion when "z" equals one, rather than the mass of an electrically neutral molecule. Please double-check the HRMS data to ensure accurate representation and interpretation of the mass spectrometry results.

Answer :

We do not fully understand the concern the referee wants to raise. We are well aware of the definition of the term "m/z" and the fact that exact mass calculations should take into account the mass of the missing electron. However, as stated in the first revision, the measured and calculated values provided by *Masslynx* software are not corrected for the mass of the electron. Thus, we respectfully disagree with the referee as we believe uncorrected measured data shall only be compared to uncorrected

calculated data, to avoid the miscalculation of the determined mass error. In the literature, both corrected and uncorrected values are found, depending on the MS manufacturer's software.

Please see : <https://doi.org/10.1007/s13361-018-2101-0>

“ When applying the same software for the calculation of the theoretical accurate mass and the MS analysis, the same mass difference will be applied for the calibration of the system and no error will be made if both the calibrant and the analyte of interest have the same charge state.”

And : <https://doi.org/10.1002/rcm.3102>

“The reason for this discrepancy is that some software packages neglect to subtract the mass of an electron in a protonated molecule $[M+H]^+$ in the positive ion mode”

As we do agree the lack of standardization of the different manufacturer's software can be confusing, we added a sentence in the SI that points out that measured and calculated mass are not corrected for the mass of the electron. HRMS data were double checked and seem accurate.

Reviewer #3 (Remarks to the Author):

The authors describe a nitrogen isotope exchange strategy by utilizing a Zincke imine intermediate and extend this method to include carbon isotope enrichment. However, the novelty of this work, also listed as a concern of Reviewer 2, still appears to be slightly diminished, in my opinion, due to the preprints on ChemRxiv highlighting $^{14}N \rightarrow ^{15}N$ isotope exchange strategies (doi: 10.26434/chemrxiv-2023-30dtw, doi: 10.26434/chemrxiv-2023-v4hx2, and doi: 10.26434/chemrxiv2023-cb5lg). Furthermore, the reported pyridine ring opening strategy (Tf₂O 1.0 equiv, HNBn₂ 1.2 equiv, 2,4,6-collidine 1.0 equiv, and EtOAc for 60 °C, 1.0 h), as I also mentioned in my initial review, is extremely similar to the one presented by McNally and coworkers (Tf₂O 1.0 equiv, HNBn₂ 1.2 equiv, collidine 1.0 equiv, and EtOAc for -78 °C, 30 min) in 2022 (10.1126/science.add8980). The work in the other two $^{14}N \rightarrow ^{15}N$ preprints (doi: 10.26434/chemrxiv-2023-30dtw, doi and doi: 10.26434/chemrxiv2023-cb5lg), not from McNally and coworkers, at least show a deviation in the ring opening strategy originally reported by McNally, aside from Tf₂O. The ^{13}N -labeling experiment, which is now in the SI, is a novel radiolabeling approach. However, I, as well as the other reviewers, had expressed concerns and questions on this sole example. Unfortunately, the majority of these concerns were dismissed by authors. As a result, I still believe the novelty of this body of work does not meet the requirement for Nature Communications. In my opinion, the authors should completely remove the ^{13}N -labeling experiment, and expand on this radiolabeling strategy and scope for a high impact publication. The remaining work should be more suitable for JACS which is where all the above preprints (doi: 10.26434/chemrxiv-2023-30dtw, doi: 10.26434/chemrxiv-2023-v4hx2, and doi: 10.26434/chemrxiv2023-cb5lg) were ultimately published.

Answer :

[Redacted: The ^{13}N labeling experiments were kept in the SI as proof-of-concept study and as part of an editorial decision]